Transcriptomic analysis identifies genes and pathways related to myrmecophagy in the Malayan pangolin (Manis javanica)

Ma Jing-E 1
Li Lin-Miao 1
Jiang Hai-Ying 1
Zhang Xiu-Juan 1
Li Juan 1
Li Guan-Yu 1
Yuan Li-Hong 1
Wu Jun 2
Chen Jin-Ping chenjp@gdei.gd.cn chenjp@giabr.gd.cn 1
1 Guangdong Key Laboratory of Animal Conservation and Resource, Guangdong Public Laboratory of Wild Animal Conservation and Utilization, Guangdong Institute of Applied Biological Resources , Guangzhou , Guangdong , China
2 Wildlife Disease Surveillance and Molecular Ecology Research Center, Nanjing Institute of Environmental Sciences under Ministry of Environmental Protection , Nanjing , Jiangsu , China
Braun Edward
Electronic publication date: 2017 Dec 22
Publication date: 2017
Volume: 5
Electronic Location ID: e4140
Received 2017 Jul 31; Accepted 2017 Nov 15
Copyright: ©2017 Ma et al.
Copyright year: 2017
Copyright holder: Ma et al.
License: This is an open access article distributed under the terms of the Creative Commons Attribution License, which permits unrestricted use, distribution, reproduction and adaptation in any medium and for any purpose provided that it is properly attributed. For attribution, the original author(s), title, publication source (PeerJ) and either DOI or URL of the article must be cited.
License URL: https://creativecommons.org/licenses/by/4.0/

Keywords: Digestion, Myrmecophagy, Conservation, Pangolin

Funding: Planning Funds of Scientific and Technological of Guangdong Province 2016B070701016 2017A050503005 Training Fund of Guangdong Institute of Applied Biological Resources for Postdoctoral Researchers NO.GIABR-pyjj201601 Funds for Environment Construction and Capacity Building of GDAS’ Research Platform 2016GDASPT-0107 GDAS Special Project of Science and Technology Development 2017GDASCX-0107 Natural Science Foundation of Guangdong Province, China 2017A030310425 This project was supported by the Planning Funds of Scientific and Technological of Guangdong Province (2016B070701016 and 2017A050503005) and the Training Fund of Guangdong Institute of Applied Biological Resources for Postdoctoral Researchers (NO.GIABR-pyjj201601), the Funds for Environment Construction and Capacity Building of GDAS’ Research Platform (2016GDASPT-0107), GDAS Special Project of Science and Technology Development (2017GDASCX-0107), and the Natural Science Foundation of Guangdong Province, China (2017A030310425). There was no additional external funding received for this study. The funders had no role in study design, data collection and analysis, decision to publish, or preparation of the manuscript.

==============================
The Malayan pangolin (Manis javanica) is an unusual, scale-covered, toothless mammal that specializes in myrmecophagy. Due to their threatened status and continuing decline in the wild, concerted efforts have been made to conserve and rescue this species in captivity in China. Maintaining this species in captivity is a significant challenge, partly because little is known of the molecular mechanisms of its digestive system. Here, the first large-scale sequencing analyses of the salivary gland, liver and small intestine transcriptomes of an adult M. javanica genome were performed, and the results were compared with published liver transcriptome profiles for a pregnant M. javanica female. A total of 24,452 transcripts were obtained, among which 22,538 were annotated on the basis of seven databases. In addition, 3,373 new genes were predicted, of which 1,459 were annotated. Several pathways were found to be involved in myrmecophagy, including olfactory transduction, amino sugar and nucleotide sugar metabolism, lipid metabolism, and terpenoid and polyketide metabolism pathways. Many of the annotated transcripts were involved in digestive functions: 997 transcripts were related to sensory perception, 129 were related to digestive enzyme gene families, and 199 were related to molecular transporters. One transcript for an acidic mammalian chitinase was found in the annotated data, and this might be closely related to the unique digestive function of pangolins. These pathways and transcripts are involved in specialization processes related to myrmecophagy (a form of insectivory) and carbohydrate, protein and lipid digestive pathways, probably reflecting adaptations to myrmecophagy. Our study is the first to investigate the molecular mechanisms underlying myrmecophagy in M. javanica, and we hope that our results may play a role in the conservation of this species.

Introduction

Pangolins, also known as scaly anteaters, are eutherians and placental mammals. Eight pangolin species are recognized: four from Asia, Manis. javanica, M. pentadactyla, M. crassicaudata, and M. culionensis, and four from Africa, Phataginus. tricuspis, P. tetradactyla, Smutsia. gigantea, and S. temminckii, the genus Manis to the four Asian species while assigning the two African tree pangolin species to the genus Phataginus and the two terrestrial African pangolin species to the genus Smutsia (Du Toit Z et al., 2014; Gaudin, Emry & Wible, 2009). M. javanica is found mainly in Southeast Asia (Trageser et al., 2017). In pangolins, unlike other placental mammals, the skin is covered by large and overlapping keratinized scales (Meyer et al., 2013). Furthermore, pangolins are edentulous or toothless and are thus specialized within an already unusual mammalian dietary niche (Pietersen et al., 2016; Yang et al., 2007). Pangolins feed mainly on ants and termites (Ashokkumar et al., 2017; Pietersen et al., 2016; Mahmood et al., 2013; Richer, Coulson & Heath, 1997; Swart, Richardson & Ferguson, 1999). Pangolins also have a well-developed muscular system for fossorial or arboreal behavior and a remarkable olfactory system (Choo et al., 2016). As predators of ants and termites, pangolins have a specialized diet and perform an important ecological role in regulating insect populations (Hua et al., 2015). Individual adult pangolins have been estimated to consume more than 70 million insects annually and pangolins provide an important regulatory function in controlling both ants and termites (Hua et al., 2015; Pietersen et al., 2016; Yang et al., 2007). In addition to their ecological value, pangolins are extremely economically important animals. Pangolins are the most poached and trafficked mammal in the world because of the high demand for their meat, which is considered a delicacy, and their scales are used in traditional medicine (Trageser et al., 2017).

Manis. javanica is classified as critically endangered according to the International Union for Conservation of Nature (IUCN) Red List of Threatened Species® (Challender et al., 2014) and has been included in Appendix I of the Convention on International Trade in Endangered Species of Wild Fauna and Flora (CITES I). The numbers of M. javanica in the wild have been dramatically declining for several reasons. A major threat is the rapid loss and deterioration of their natural habitat as a result of deforestation; illegal hunting, which might reflect the pangolin’s economic value; and human agricultural expansion (Hua et al., 2015). Therefore, artificial breeding may be the best choice for ensuring pangolin survival. A suitable artificial diet is one of the critical limiting factors in raising pangolins in captivity because they can have many digestive problems (Hua et al., 2015). Pangolins have adapted to a highly specialized diet of ants and termites; thus it is difficult to replace their natural food completely by artificial food (Hua et al., 2015; Yang et al., 2007). Pangolins favor high-protein, high-fat, and high-calorie food, and they have the notable ability to digest and absorb chitin in the digestive system (Hua et al., 2015; Yang et al., 2007). Chitin is a linear polymer of N-acetyl-beta-glucosaminidase subunits linked by a beta-1, 4 glycosidic bond and is one of the constituents of insect exoskeletons and the peritrophic membranes of ants (Kawashima et al., 2016). The internal degradation of chitin particles is mainly catalyzed by chitinase (Strobel et al., 2013).

Over the past 150 years, several zoos have tried to maintain pangolins. However, because of inadequate diets, these animals have not been successfully maintained for long periods by most zoos (Hua et al., 2015). Some formulas for diets fed to M. javanica in captivity use a paste mixture of several kinds of food, such as hard-boiled chicken eggs, multivitamin liquid, horse meat, water, mealworms, insectivore pellets, salmon oil, and powdered termite mound (Vijayan, Yeong & Ling, 2009). Digestive disorders often appear in pangolins fed with artificial food, and the faeces of the animals become fluid. Several researchers have suggested that a certain proportion of chitin might be the key to artificial diets for pangolins (Ya-yong et al., 1999; Yang et al., 2007); however, an understanding of the molecular genetics that might provide a theoretical basis for raising M. javanica in captivity is lacking.

Genetic studies of endangered species have become increasingly widespread in the last two years (Choo et al., 2016; Mwale et al., 2017; Yusoff et al., 2016; Zhihai et al., 2016). In particular, the genomes and transcriptomes of M. javanica have been reported (Choo et al., 2016; Yusoff et al., 2016). Currently, increasing amounts of molecular information about M. javanica have become available as a result of the emergence of high-throughput next-generation sequencing (NGS) technologies. The complete genome sequencing of M. javanica and the transcriptome sequencing of eight organs, including the heart, liver, spleen, lung, kidney, thymus, cerebellum, and cerebrum, have progressively revealed unknown aspects of pangolin biology (Yusoff et al., 2016). High-quality transcriptomes have been used for analyses of the functional and phylogenetic aspects of immunity biology (Yusoff et al., 2016), but genetic research regarding myrmecophagy is still lacking. Here, two questions are raised. Are molecular pathways involved in the evolution of this dietary adaption for myrmecophagy, and how do they affect the appearance of this feature? In terms of dietary adaptation, we wanted to investigate whether specific genes exist for digestive function. We selected the liver, small intestine, and salivary glands for transcriptome sequencing to analyze the genetic selection of potential candidate genes involved in myrmecophagy. Salivary glands are important amylase secretory organs although liver secretions also contain amylase and lipase, and the small intestine is involved in the absorption of nutrients. We aimed to analyze the unique feeding behavior of pangolins in the hope that our results may provide a new approach to the protection of this species.

Materials and Methods

Ethics statement

All animal procedures were approved by the ethics committee for animal experiments at the Guangdong Institute of Applied Biological Resources (reference number: G2ABR20170523), and followed basic principles.

Biological sample

Briefly, an adult female M. javanica was provided by the Dongguan Institute of Qingfengyuan Animal Medicine (Dongguan, Guangdong, China). The specimen was a wild individual that died in the process of rescue; the animal was 4.08 kg in weight before its natural death. The animal was dissected immediately after its natural death. During the dissection, ants were found in its stomach. The salivary glands, liver, and small intestine were collected as soon as possible, frozen in liquid nitrogen, and stored at −80 °C until RNA extraction.

RNA isolation, cDNA library construction and Illumina sequencing

Total RNA was extracted from the three tissues using RNAiso reagent (Takara, Otsu, Japan) and was treated with DNase I (Takara). RNA purity was checked using a NanoPhotometer spectrophotometer (IMPLEN, CA, USA). RNA concentration was measured using the Qubit RNA Assay Kit and a Qubit 2.0 Flurometer (Life Technologies, Carlsbad, CA, USA), and RNA integrity was assessed using the RNA Nano 6000 Assay Kit and the Agilent Bioanalyzer 2100 system (Agilent Technologies, Santa Clara, CA, USA). RNA was frozen at −80 °C until cDNA library construction.

RNA samples were then mixed with fragmentation buffer (Ambion, Foster City, CA, USA) and fragmented into short fragments; the average insert size was 200 bp. mRNA was purified from total RNA using poly-T oligo-attached magnetic beads. DNA contaminants were further removed through DNase enzyme digestion followed by rRNA removal. cDNA synthesis was then followed by PCR amplification to generate a complete cDNA library, which was sent for sequencing in a flowcell on the Illumina HiSeq™ 2000 platform using the TruSeq PE Cluster Kit V3-cBot-HS (Illumina PE-401-3001) and TruSeq SBS Kit-HS v3 200 cycles (Illumina FC-401-3001).

Data assembly and annotation

Four groups of sequencing data were used for assembly and annotation and the data were annotated using a published paper housed at the National Center for Biotechnology Information (NCBI) Sequence Read Archive (SRA) accession number SRR2561213 (Yusoff et al., 2016). The primary sequencing data were cleaned by (a) removing reads with adaptors; (b) removing reads where the proportion of unknown bases (N) exceeded 5% (−n = 0.05); and (c) removing low-quality reads (reads in which the proportion of bases with quality <10 exceeded 20%, i.e., −l = 10 and −q > 0.2). Clean reads were aligned to reference sequences using a spliced read mapper for RNA-Seq-TopHat2 (http://ccb.jhu.edu/software/tophat/index.shtml), which is based on Bowtie2 (http://bowtie.cbcb.umd.edu/). The parameter values of RNA-Seq-TopHat2 were as follows; –read-mismatches: 2, –read-edit-dist: 2, –max-intron-length: 5000000, –library-type: fr-unstranded, –GTF: genome.gtf, –mate-inner-dist: 40, –solexa1.3-quals, and other parameters were set to the default values. There were three steps: (a) align the reads to the reference transcriptomes; (b) align the reads to the reference genome without the alignment described in (a); and (c) align the reads segmentally to the reference genome without the alignment described in (b). Following this, the alignment data were used to calculate the distribution and coverage of the reads on the reference genes.

To offer complementary insights, all transcripts (>200 bp) were annotated on the basis of basic local alignment search tool (BLASTX) results with e-values of 1e−5 against a total of seven databases, including the non-redundant protein database (NR, ftp://ftp.ncbi.nih.gov/blast/db/); a manually annotated, non-redundant protein sequence database (Swiss-Prot, http://www.uniprot.org/); the Translated EMBL Nucleotide Sequence Data Library (TrEMBL, http://www.bioinfo.pte.hu/more/TrEMBL.htm), which complements the Swiss-Prot protein knowledgebase; the Kyoto Encyclopedia of Genes and Genomes (KEGG, http://www.genome.jp/kegg/), which is used to understand the high-level functions and utilities of biological system (such as the cell, the organism and the ecosystem) based on molecular-level information, especially for large-scale molecular datasets generated by genome sequencing and other high-throughput experimental technologies; Gene Ontology (GO, http://www.geneontology.org/), which described biological processes, molecular functions, and cellular components; and Clusters of Orthologous Groups of proteins (KOG/COG: COG, http://www.ncbi.nlm.nih.gov/COG/, KOG, http://www.ncbi.nlm.nih.gov/KOG/). The best-aligned results were used to determine the sequence direction of the transcripts. If the results from the different databases were in conflict, the sequence direction of the transcript was determined using the prioritization order NR, Swiss-Prot, KEGG and COG.

Analysis of differentially expressed genes

Gene expression levels were estimated based on the fragments per kilo base (kb) of transcript per million fragments mapped. The formula is as follows: FPKM = cDNA Fragments/Mapped Fragments (Millions)/Transcript Length (kb). Prior to differential gene expression analysis, for each sequenced library, the read counts were adjusted using the edgeR program package and a scaling normalized factor (Robinson, McCarthy & Smyth, 2010). The differential expression analysis of two samples was performed using the DEGseq R package (Wang et al., 2010). P-value was adjusted using q value, and q value < 0.005 & |log2 (fold change)| ≥ 1 was set as the threshold for significantly differential expression (Storey & Tibshirani, 2003).

Correlation between any two pangolin tissue transcriptomes

To examine the similarity between the different organ transcriptomes, the expression levels of the transcripts (FPKM) in the transcriptomes of each tissue were manipulated using the tool “RSEM-calculate-expression” in the RSEM pipeline (http://deweylab.biostat.wisc.edu/rsem/README.html), which performs accurate transcript quantification from RNA-Seq data with or without a reference genome. The reads for each tissue were mapped to the transcripts (Li & Dewey, 2011). Gene expression values, expressed as log 10 (FPKM + 1) for the transcriptomic data from each tissue, were plotted against one another to produce scatter plots. R2 values were then calculated from the scatter plots to assess the correlation between any two M. javanica transcriptomes.

Results

Illumina sequencing and assembly

To obtain a comprehensive and representative transcriptome of M. javanica, 97,353,658 high-quality clean reads (for a total length of 2,920,609,740 bp) were generated from the three tissues after the removal of the adaptor sequences. All high-quality sequencing reads from M. javanica are available on the NCBI Gene Expression Omnibus (GEO) database under accession numbers GSM2667949, GSM2667950 and GSM2667951. The average proportion of high-quality clean reads was 95.58% (Table 1). Clean reads were assembled into long assembled sequences (contigs) using TopHat2. The alignment efficiency between the sample and reference genome ranged from 69.57–89.30% (Table 2), while the alignment efficiency between the sample and the exons of the reference genome ranged from 76.11–85.16% (Fig. S1).

Table 1 Sequencing statistics for the transcriptomes obtained from the salivary glands, liver, and small intestine of adult female Malayan pangolin (Manis javanica).

Organ	Raw pairs (bp)	Clean pairs (bp)	Clean bases (bp)	GC content	% ≥Q30%	
Small intestine	20,296,915	15,117,014	4,535,104,200	53.71%	95.30%	
Liver	33,288,751	22,073,184	6,621,955,200	53.15%	94.67%	
Salivary glands	20,297,074	14,857,037	4,457,111,100	51.33%	95.58%	
Referred liver	–	45,306,423	9,061,284,600	50.87%	90.35%	

Table 2 Summary of the sequencing data aligned to the Malayan pangolin (Manis javanica) whole genome reference sequence.

Organ	Total reads	Mapped reads	Unique mapped reads	Multiple map reads	
Small intestine	30,234,028	22,277,733
(73.68%)	21,843,354
(72.25%)	434,379
(1.44%)	
Liver	44,146,368	31,965,251
(72.41%)	31,694,559
(71.79%)	270,692
(0.61%)	
Salivary glands	29,714,074	20,673,075
(69.57%)	19,911,660
(67.01%)	761,415
(2.56%)	
Reference liver	90,612,846	80,913,231
(89.30%)	80,064,920
(88.36%)	848,311
(0.94%)	

Functional annotation

From the M. javanica transcriptome, 22,538 transcripts (93.05%) were annotated on the basis of the COG, GO, KEGG, KOG, Swiss-Prot, TrEMBL, and NR databases using BLAST. A total of 6,228 transcripts were annotated against the COG database, followed by 13,977, 14,115, 16,648, 17,135, and 20,964 transcripts annotated on the basis of the KEGG, KOG, GO, Swiss-Prot, and TrEMBL databases, respectively (Table S1). As expected, the majority of the 22,473 transcripts matched the NR databases (e-value <10−5) (Fig. S2 and File S1). The M. javanica transcripts were annotated on the basis of the top BLASTX hits in the species distribution statistics. The top five organisms were Ceratotherium simum (2,382 transcripts, 10.60%), Equus caballus (1,402, 6.24%), Canis lupus (1,349, 6.01%), Mustela putorius (1,044, 4.65%) and Odobenus rosmarus (1,022, 4.55%) (Fig. S3).

COG and KOG analysis

In the COG database, the largest category of M. javanica annotated transcripts was general function prediction only (R) (2,438 transcripts, 27.76%), followed by replication, recombination, and repair (L) (848, 9.65%); transcription (K) (845, 9.62%); signal transduction mechanisms (T) (790, 8.99%); and post-translational modification, protein turnover, and chaperones (O) (484, 5.51%) (Fig. S4A). In the KOG database, the largest category of M. javanica annotated transcripts was general function prediction only (R) (2,910 transcripts, 18.3%), followed by signal transduction mechanisms (T) (2,744, 17.26%); post-translational modification, protein turnover, and chaperones (O) (1,259, 7.92%); function unknown (S) (1,182, 7.43%); and transcription (K) (1070, 6.73%) (Fig. S4B).

In both the COG and KOG analyses, several transcripts were involved in the transport and metabolism of the three major nutrients: 136 transcripts were related to carbohydrate transport and metabolism, 122 were related to lipid transport and metabolism, and 124 were related to amino acid transport and metabolism (File S2).

Gene Ontology (GO)

Annotation of the M. javanica transcripts with the GO database classified 16,649 transcripts into 61 small classes in three ontologies: biological processes, molecular functions, and cellular components. A total of 44.36% of the transcripts were assigned to biological processes, 16.26% to molecular functions, and 39.38% to cellular components.

In the biological process ontology, the most highly represented terms were cellular processes (10,494, 63.03%), single-organism processes (9,554, 57.38%), and biological regulation (7,986, 47.97%). The fourth most represented term was metabolic process (7,173, 43.08%), which was followed by response to stimulus (4,915, 29.52%), multicellular organismal process (3,644, 21.89%), signaling (3,161, 18.99%), localization (3,001, 18.03%), developmental process (2,890, 17.36%), and cellular component organization or biogenesis (2,667, 16.02%). The terms associated with biological regulation and metabolic process might be indicative of M. javanica transcriptome involvement in various digestive activities.

For molecular functions, the sequences were mainly assigned to binding (9,371, 56.29%) and catalytic activity (5,347, 32.12%) followed by molecular transducer activity (1,750, 10.51%), receptor activity (1,681, 10.1%) and transporter activity (1,026, 6.16%), which might be involved in food digestion and absorption.

As anticipated, cell part (11,445, 68.74%) and cell (11,412, 68.54%) were the predominant terms assigned to the pangolin transcriptome in cellular components followed by organelle (8,014, 48.14%), membrane (5,980, 35.92%), membrane part (4,441, 26.67%), organelle part (4,079, 24.5%), macromolecular complex (3,413, 20.5%) and extracellular region (1,044, 6.27%) (Fig. S5, File S3). Overall, these results indicate that a broad range of biological activities were related to the expressed pangolin transcriptome, representing a pooled collection of the three digestive tissues sequenced.

KEGG pathway analysis

To identify the pathways in which the M. javanica transcripts were involved, the transcripts were mapped on the basis of KEGG pathways. A total of 13,977 (57.71%) M. javanica transcripts were associated with 290 unique KEGG pathways, with a total of 15 unique KEGG pathways representing cellular processes, followed by 22, 27, 66, 68, and 89 representing genetic information passing, environmental information processing, organismal systems, human diseases, and metabolism, respectively (File S4).

The most-represented pathways in the M. javanica transcripts included olfactory transduction (969 transcripts) and pathways in cancer (444 transcripts), followed by the PI3K-Akt signaling pathway (391 transcripts), the MAPK signaling pathway (300 transcripts), and neuroactive ligand–receptor interaction (292 transcripts) (Fig. S6). The sense of smell is closely related to the biological activity of instinctive behaviors such as feeding, and the olfactory pathway plays a key role in the specific recognition of smells, thus leading the animal to different foods. A total of 969 genes were associated with olfactory transduction in the M. javanica transcripts, 942 transcripts of which were annotated as various kinds of olfactory receptors. These findings may explain the keen sense of smell in M. javanica.

Metabolic pathway analysis

A total of 1,814 transcripts were associated with 89 unique KEGG metabolic pathways. Most transcripts were involved in lipid metabolism (431 transcripts), carbohydrate metabolism (365 transcripts), amino acid metabolism (321 transcripts), glycan biosynthesis and metabolism (271 transcripts), nucleotide metabolism (233 transcripts), and the metabolism of cofactors and vitamins (225 transcripts). Other transcripts were associated with global and overview maps (199 transcripts), energy metabolism (167 transcripts), the metabolism of other amino acids (119 transcripts), and xenobiotic biodegradation and metabolism (116 transcripts); a small number of the transcripts were associated with terpenoid and polyketide metabolism (27 transcripts) and secondary metabolite biosynthesis (13 transcripts) (Fig. 1A).

Figure 1 The metabolic pathway analysis of transcripts from Manis javanica.

(A) Metabolism, (B) Carbohydrate metabolism, (C) Lipid metabolism, (D) Amino acid metabolism, (E) Cofactors and vitamin metabolism and terpenoid and polyketide metabolism. The x-axis shows the numbers of annotated transcripts in one class, and the y-axis shows the KEGG function classes.

Carbohydrate metabolism

Inositol phosphate metabolism (73 transcripts), glycolysis/gluconeogenesis (70 transcripts), starch and sucrose metabolism (57 transcripts), and amino sugar and nucleotide sugar metabolism (53 transcripts) were at the top of the carbohydrate metabolic lists, whereas ascorbate and aldarate metabolism were at the bottom.

The chitin-degrading enzyme acidic mammalian chitinase (CHIA), which is involved in the degradation of the chitin in the insect cuticle and the peritrophic membrane of the dietary ant, was found in the amino sugar and nucleotide sugar metabolism pathway (KEGG: 00520), thus suggesting that this pathway may be directly involved in ant digestion by M. javanica (Fig. 1B).

Lipid metabolism

Glycerophospholipid metabolism (100 transcripts), arachidonic acid metabolism (76 transcripts), steroid hormone biosynthesis (68 transcripts), and sphingolipid metabolism (55 transcripts) were at the top of the lipid metabolism list; in contrast, fatty acid biosynthesis (55 transcripts) was at the bottom. We identified transcripts from several pathways in unsaturated fatty acid metabolism, including arachidonic acid metabolism (76 transcripts), linoleic acid metabolism (36 transcripts), alpha-linolenic acid metabolism pathways (23 transcripts), and the biosynthesis of unsaturated fatty acids (23 transcripts) (Fig. 1C).

Amino acid metabolism

Lysine degradation (66 transcripts), valine, leucine and isoleucine degradation (65 transcripts), arginine and proline metabolism (64 transcripts), and tryptophan metabolism (35 transcripts) were at the top of the amino acid metabolic lists. The biosynthesis pathways of some amino acids, such as phenylalanine, tyrosine, and tryptophan (6 transcripts); valine, leucine and isoleucine (5 transcripts); and lysine (2 transcripts) were at the bottom (Fig. 1D). None of the transcripts were found to be involved in arginine biosynthesis.

Cofactors and vitamin metabolism and terpenoid and polyketide metabolism

Retinol metabolism (65 transcripts), porphyrin and chlorophyll metabolism (46 transcripts), nicotinate and nicotinamide metabolism (36 transcripts), and pantothenate and CoA biosynthesis (31 transcripts) were at the top of the cofactors and vitamin metabolism lists. There was only one list relating to terpenoids and polyketides; terpenoid backbone biosynthesis (27 transcripts) (Figs. 1E and 2).

Figure 2 Terpenoid backbone biosynthesis (KEGG map 00900).

Annotation of the new transcripts

On the basis of the genome sequences of M. javanica, Cufflinks software was used to join the mapped reads, to compare them with the annotated information for the original genome, and to search for the gapped sequences, which were not annotated. A total of 3,373 new transcripts were discovered (File S5), of which 1,459 transcripts were annotated on the basis of the COG, GO, KEGG, KOG, Swiss-Prot, TrEMBL and NR databases using BLAST (Table S2 and File S6).

In the Gene Ontology analysis, 75 new transcripts were involved in 22 metabolic categories. Some of the new genes were involved in inositol metabolism (GO: 0006020; e.g., Manis_javanica_newGene_958), and some were involved in the linoleic acid metabolism (GO: 0043651; e.g., Manis_javanica_newGene_12722). For KEGG, 114 new genes were related to metabolic function, including lipid metabolism (41 transcripts), carbohydrate metabolism (24 transcripts), cofactor and vitamin metabolism (22 transcripts), and amino acid metabolism (20 transcripts) (Fig. S7).

Table 3 Summary of genes related to the diet of the Malayan pangolin (Manis javanica).

Type	Gene name	
Opsin	GRK1, OPN1SW, OPN1LW, OPN4, PDE6D, PDE6G, PDE6H, RHO	
Taste	TAS1R2, TAS1R3, TAS2R1, TAS2R4, TAS2R7, TAS2R10, TAS2R30, TAS2R38, TAS2R40	
Olfactory	CNGA2, DTMT, OLF1, OLF2, OLF3, OLF4, OR1A1, OR1D2, OR1E1, OR1E2, OR1E5, OR1G1, OR3A1, OR3A2, OR3A3	
Carbohydrases	AGL, AMY2, CHIA, CHI3L1, CHID1, GAA, GANAB, GANC, GBA3, GLB1, GLB1L, PRKCSH, SI	
Lipases	ABHD6, ABHD12, CEL, CLPS, DDHD1, GPLD1, Group XV phospholipase A2, LIPA, LIPC, LIPE, LIPF, LIPH, LMF1, LMF2, LPL, LYPLAL1, NAPEPLD, PLA1A, PLA2G1B, PLA2G2A, PLA2G3, PLA2G4A, PLA2R1, PLB1, PLBD1, PLBD2, PLD3, PNLIP, PNLIPRP1, PNLIPRP2, PNPLA2, PNPLA8	
Protease	Anionic trypsin, ANPEP, Cationic trypsin, CELA1, Chymotrypsin A chain C, CTRB1, CTRC, DPP6, DNPEP, ENPEP, ERAP2, LAP3, METAP1, METAP2, NPEPL1, PGC, PRSS12, Trypsin, XPNPEP1, XPNPEP2, XPNPEP3	
Transporters	SLC1A1, SLC1A3, SLC1A6, SLC1A4, SLC1A5, SLC7A8, SLC43A2, SLC6A15, SLC6A17, SLC6A19, SLC38A1, SLC38A2, SLC38A4, SLC38A5, SLC38A7, SLC38A10, SLC38A11, SLC7A2, SLC7A14, SLC7A11, SLC25A29, SLC2A1, SLC2A2, SLC2A3, SLC2A4, SLC2A5, SLC2A8, SLC2A9, SLC2A12, SLC35A4, SLC35A5, SLC50A1, SLC35A3, SLC35B4, SLC35D2, CLCN3, CLCN5, CLCN7, MFSD5, MAGT1, MMGT1, MRS2, NIPA2, NIPAL1, Sodium-independent sulfate anion transporter, SLC4A4, SLC20A1, SLC20A2, SLCO1C1, SLCO3A1, SLCO4C1, LMBRD1, SLC5A6, SLC19A3, SLC25A32, SLC52A2, SLC52A3, SLC5A1, SLC5A4, SLC5A10, SLC5A2, SLC28A1, SLC5A3, APOA1, APOA2, APOB, Apolipoprotein A-IV, APOC2, APOC3, APOC4, APOD, APOE, APOM, APOO, SLC6A2, SLC6A3, SLC6A4, SLC6A8, SLC6A9, SLC6A12, SLC6A13, SLC10A2, SLC5A12, SLC16A1, SLC16A9, SLC16A13, SLC17A6, SLC17A7, SLC26A2, SLC29A3, SLC44A2, SLC44A3, SLC44A4, SLC44A5, SLC45A2, SLC46A2	

Gene expression repertoire

Distributions of potential transcripts related to feeding among the three tissue libraries are shown in Table 3, Table S3 and File S7, including the 997 transcripts related to sensory perception. Of these transcripts, 972 were related to olfaction, and 11 and 14 transcripts were related to vision and taste, respectively. A total of 133 transcripts were related to digestive enzyme gene families; 70 of these transcripts were related to lipid degradation, and 39 and 20 were related to the degradation of proteins and carbohydrates, respectively. These genes were considered to be involved in the profile of food choice, digestion and absorption, which might serve as a molecular mechanism in myrmecophagy. Among these transcripts, acidic mammalian chitinase (CHIA), chitinase-3-like protein 1 (CHI3L1), and chitinase domain-containing protein 1 (CHID1) were related to chitin degradation. A total of 199 transcripts were related to molecular transporters, including sugar transporters, amino acid transporters, apolipoprotein transporters, cationic/anion transporters, vitamin transporters, cotransporters and others; among these, the UDP-N-acetylglucosamine transporter (SLC35A3, SLC35B4, and SLC35D2) was related to the decomposer absorption of the chitin unit N-acetylglucosamine.

Figure 3 Pairwise correlation between organs (A–F).

Pairwise comparisons of different transcriptomic profiles

To examine the similarity among organ transcriptomes, we performed statistical correlation analysis for each pair of organs using log10 (FPKM + 1) transformation to normalize the plots (Fig. 3). Our data showed that two liver transcriptome expression profiles were the most similar (R2 = 0.53), followed by those of the liver and small intestine (R2 = 0.30). The salivary glands and small intestine had the least similar transcriptomic profiles (R2 = 2e−0.4). The low correlation between the liver transcriptomes found in our study and that of Yusoff et al. (2016) may reflect the varying complexity between the same organs in different individuals, possibly because one of the two specimens was pregnant. The differences between each pair of compared organs reflect the different digestive functions of the three organs.

A total of 11,055 transcripts were expressed (FPKM >1.0) in all three tissues; of these, 3,947 were annotated in KEGG. Several transcripts were enriched in lipid metabolism (214 transcripts), carbohydrate metabolism (240 transcripts), and amino acid metabolism (188 transcripts) (Fig. 4). Other transcripts near the top of the metabolic lists included glycerophospholipid metabolism (62 transcripts); valine, leucine and isoleucine degradation (53 transcripts); lysine degradation (49 transcripts); and inositol phosphate metabolism (45 transcripts). The biosynthesis pathways of valine, leucine, and isoleucine (one transcript) and lysine (two transcripts) were less commonly represented.

Figure 4 The transcripts related to metabolic pathways expressed in all three tissues.

The x-axis shows the number of transcripts with the KEGG function class, which was shown above the column, and the y-axis shows the KEGG function classes.

Differentially expressed genes were identified among systems in the metabolic pathways of sugars, lipids, and amino acids. A total of 382 transcripts were differentially expressed between the small intestine and liver; this value compares with 258 transcripts that were differentially expressed between the salivary glands and small intestine. Twenty-three and 27 genes differed between the small intestine and liver for starch and sucrose metabolism and arginine and proline metabolism, respectively, and 31 genes for steroid hormone biosynthesis differed between the liver and the referred liver from (Yusoff et al., 2016) (Fig. S8).

Several transcripts were specifically expressed in a single sample, including 21 transcripts in the small intestine that were involved in 19 pathways, five transcripts in the liver that were involved in 11 pathways, 36 transcripts in the referred liver that were involved in 27 pathways, and six transcripts in the salivary glands that were involved in 10 pathways (Table S4). The highest number of specific transcripts was eight, and these transcripts were involved in the arachidonic acid metabolism in the liver reported in the previous study (Yusoff et al., 2016); six transcripts were involved in glycerophospholipid metabolism, in either lipid metabolism or arachidonic acid metabolism of the small intestine (Fig. 5).

Figure 5 The numbers of transcripts related to metabolic pathways that are specifically expressed in the salivary glands, liver, and small intestine of adult female Malayan pangolin (Manis javanica).

The x-axis shows the KEGG function classes, and the y-axis shows the number of transcripts with the corresponding KEGG function class.

Discussion and Conclusion

Based on these results, a total of 3,373 new genes were discovered in the transcriptomic datasets, of which 1,459 were annotated, and 75 new genes were involved in metabolism. The new genes provide new information for studying the myrmecophagous mechanisms of pangolins. A large number of genes were expressed in all three tissues, and several specific genes in the three systems played different roles in the metabolism of sugars, lipids, and amino acids. The digestive functions of the small intestine and salivary glands were similar in comparison with the differences between the small intestine and liver. The functions of the livers from the two different individuals differed in lipid metabolism pathways, suggesting that the ratio of lipid in the feed might be changed appropriately during pregnancy. Pangolin scales are considered an effective Chinese medicine and are thought to promote blood circulation, accelerate milk secretion, detumescence and apocenosis (Hua et al., 2015). The scales are helpful for the treatment of such human diseases. Interestingly, our results indicated that several KEGG pathways were related to human diseases, possibly indicating the mechanisms by which pangolin scales act on human diseases. However, further data are needed to elucidate the underlying mechanisms.

Ants are a protein-rich food (Tomotake, Katagiri & Yamato, 2010). They contain more than 50% crude protein, according to a nutritional value evaluation, and contain more than 20 amino acids; various microelements; special chemicals, such as formic acid and herbaceous acetaldehyde, which are triterpenoid compounds, and several vitamins (Pattarayingsakul et al., 2017). Many pangolin genes are likely to be involved in the digestion of these materials because 27 of the transcripts related to the terpenoid backbone biosynthesis, and this might be one of the biological basis for their adaptions to myrmecophagy. However, no genes were found to participate in the arginine synthesis pathway according to the KEGG analysis, and only two transcripts were involved in lysine synthesis. We did find a few transcripts encoding proteins that are necessary for the biosynthesis of amino acids that are essential in humans, such as lysine, valine, leucine, isoleucine, phenylalanine, and tryptophan (Galili, Amir & Fernie, 2016; Zhenyukh et al., 2017). These results suggest that M. javanica might have at least a limited ability to synthesize those amino acids and that arginine might be the first limiting amino acid. Therefore, it might not be necessary to add the amino acids that pangolins have the potential to synthesis to manufactured feed.. However, the regulation of the relevant transcripts is not fully characterized and the ability of pangolins to flourish without those amino acids is unclear. This issue needs further research, which might be very useful for theoretical and practical purposes.

In our M. javanica transcriptomic analysis, the same enriched GO terms were found to be enriched in biological processes in response to a stimulus as those found by Yusoff et al. (2016), indicating that the M. javanica transcripts may be actively involved in stimulus or stress responses. However, several GO terms involving biological processes were found to be enriched in our study, including metabolic and developmental processes that might be involved in food digestion and absorption. Regarding molecular functions, receptor activity was similarly well enriched. In the previous study, the other main GO term found was cytoskeletal protein binding, and this likely required the evolution of a sophisticated musculoskeletal system or the formation of pangolin scales, which are made of keratin (Yusoff et al., 2016). In our study, some enriched GO terms were also related to metabolism, including catalytic activity, molecular transducer activity and transporter activity. The observed differences might be related to the different tissues used to obtain the transcriptome data.

Chitin, one of the main components of the epidermis of ants and termites, comprises of N-acetyl-D-glucosamines (GluNAc) connected by a β-1, 4 glycosidic bond. Chitin can be digested only with chitinase and acidic mammalian chitinase (AMCase); AMCase is widely found in the digestive organs of animals (Eurich et al., 2009; Krykbaev et al., 2010; Strobel et al., 2013). The origin of the digestive processes might be closely related to the activity of AMCase, which determines the beginning of chitin. The transporter genes of UDP-N-acetyl glucosamine (SLC35A3, SLC35B4, and SLC35D2) might be directly related to the absorption of carbohydrate units during chitin metabolism (Gerardy-Schahn, Oelmann & Bakker, 2001). Therefore, chitin could be added to the formulated diets to aid in the digestion and absorption of nutrients. This suggestion is consistent with the referenced formulas mentioned in Vijayan, Yeong & Ling (2009).

Molecular adaptations consistent with the diet have also been shown in other studies. The tiger genome is particularly enriched in olfactory receptor activity, the G-protein-coupled receptor signaling pathway, signal transducer activity, amino-acid transport and protein metabolism. These markers of amino-acid metabolism have been associated with an obligatory carnivorous diet (Cho et al., 2013). We also found highly enriched transcripts for olfactory receptor activity in M. javanica. In red panda, regarding gene ontology and KEGG enriched analyses, significant terms and pathways were involved in limb development and nutrient utilization, including appendage and limb development, cilium assembly, protein digestion and absorption, and retinol metabolism; these pathways might be related to their bamboo diet (Hu et al., 2017). In the panda, loss-of-function is observed for TAS1R1, and this might prevent the panda from expressing a functional umami taste receptor, thus partly explaining why the panda diet is primarily herbivorous despite its taxonomic classification as a carnivore (Li et al., 2010). The polar bear is adapted to cope with a diet rich in fatty acids. Gene Ontology analysis for putative genes under positive selection in the polar bear lineage has shown that genes associated with adipose tissue development are enriched, reflecting the crucial role that lipids play in the ecology and life history of polar bears (Liu et al., 2014). However, transcripts involved in lipid metabolism were enriched in M. javanica, a finding that is generally consistent with the Gene Ontology analysis of the polar bear genome, while the markers of amino-acid metabolism are apparently associated with the diet in pandas and tigers.

Malayan pangolins are endangered mammals for which captive breeding provides an opportunity to study the molecular mechanisms of myrmecophagy. Our M. javanica transcriptomic datasets of the three representative tissues present the first attempt at uncovering the mysteries of the genetic mechanisms of digestion and reproduction in this rare and unique mammal. Here, the transcriptome sequencing of digestive organs was performed to observe the metabolic pathways and functional genes related to myrmecophagy in an attempt to understand the molecular mechanisms involved. The results may provide an important theoretical basis for the successful captive breeding of this species. The transcriptomic data for the three organs were obtained with a high degree of confidence, and the transcripts were well annotated, thus providing a genomic and molecular basis for future study of this lesser-known mammalian species. However, the experimental design was constrained by the availability of samples, thus, further studies with more biological replicates are warranted. Other major organs involved in digestion (such as tongue, stomach, and pancreas) should be examined in further study, and these pangolin-specific genes may also be required to understand the unique traits or adaptations of pangolins (as compared to other mammals). Together, these results might shed light on the special diet of M. javanica.

Supplemental Information

File S1 Transcripts functional annotation

Click here for additional data file.

File S2 Transcripts involved in the transport and metabolism of the three major nutrients in both the COG and KOG analyses

Click here for additional data file.

File S3 Functional distribution of GO annotation

Click here for additional data file.

File S4 Functional distribution of KEGG annotation

Click here for additional data file.

File S5 Sequences of the new transcripts

Click here for additional data file.

File S6 New transcripts functional annotation

Click here for additional data file.

File S7 Functional annotation of potential transcripts related to feeding

Click here for additional data file.

Figure S1 Genomic read distribution in different organs. (A) small intestine, (B) liver, (C) referred liver, (D) salivary glands

Click here for additional data file.

Figure S2 Pie for e-value of the transcripts matched the NR databases

Click here for additional data file.

Figure S3 Statistics of the non-redundant protein database (NR) annotations for various species relative to M. javanica

Click here for additional data file.

Figure S4 Cluster of Orthologous Groups (COG) and Clusters of Protein homology (KOG) function classification of transcripts from M. javanica

The x-axis shows the COG or KOG function classes, and the y-axis shows the number of transcripts in one class. The notation on the right shows the full names of the function classes.

Click here for additional data file.

Figure S5 Gene Ontology (GO) classification analysis of transcripts from M. javanica

The x-axis shows GO function classes. The right side of the y-axis shows the number of transcripts with the GO function, and the left side shows the percentage.

Click here for additional data file.

Figure S6 The known Kyoto Encyclopedia of Genes and Genomes (KEGG) pathway analysis of transcripts from M. javanica

Click here for additional data file.

Figure S7 The KEGG pathway analysis of new transcripts from the genome of M. javanica

The x-axis shows the percentage of transcripts in one class, and the y-axis shows the KEGG function classes, the number of transcripts with the KEGG function class was shown above the column.

Click here for additional data file.

Figure S8 The correlation between any two organs, resulting from pairwise comparison using the number of metabolic pathways

G, L, S and liver represent small intestine, liver, salivary glands and referred liver, respectively.

Click here for additional data file.

Table S1 Statistics of annotation results

Click here for additional data file.

Table S2 Statistic results of annotated new transcripts

Click here for additional data file.

Table S3 The full name of genes related to the diet of M. javanica

Click here for additional data file.

Table S4 The genes specifically expressed in the single tissue

Click here for additional data file.

We acknowledge BMKCloud (Beijing, China) for the data analysis. We acknowledge the Dongguan Institute of Qingfengyuan Animal Medicinal (Dongguan, Guangdong, China) for their selfless offering of three female pangolin samples.

Additional Information and Declarations

Competing Interests

Author Contributions

Animal Ethics

Data Availability

The authors declare there are no competing interests.

Jing-E Ma conceived and designed the experiments, performed the experiments, wrote the paper, prepared figures and/or tables.

Lin-Miao Li and Guan-Yu Li performed the experiments.

Hai-Ying Jiang and Juan Li analyzed the data.

Xiu-Juan Zhang contributed reagents/materials/analysis tools.

Li-Hong Yuan contributed reagents/materials/analysis tools, reviewed drafts of the paper.

Jun Wu reviewed drafts of the paper.

Jin-Ping Chen conceived and designed the experiments, wrote the paper, reviewed drafts of the paper.

The following information was supplied relating to ethical approvals (i.e., approving body and any reference numbers):

Guangdong Institute of Applied Biological Resources provided full approval for this study (reference number: G2ABR20170523).

The following information was supplied regarding data availability:

The data are available on the NCBI Gene Expression Omnibus (GEO) database with the accession: GSM2667949, GSM2667950 and GSM2667951.

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
