# Peer review of "Transcriptomic analysis identifies genes and pathways related to myrmecophagy in the Malayan pangolin (Manis javanica)"

_PeerJ, doi:10.7717/peerj.4140_

## Round 0.1 · original submission · Major Revisions

I have now received three reviews and all see substantial value in this manuscript. I agree with their assessment. There is obvious value in the publication of molecular data relevant to these fascinating and highly endangered organisms.

All of the reviewers commented on aspects of the written communication and I certainly urge the authors to take great care to address their English usage. I generally found the paper to be quite readable, but I think it is always important to be careful. The second reviewer also pointed out areas that should have received citations and provided a marked manuscript with a number of suggestions regarding those citations (and suggestions regarding the English usage). Please take care to use those mark-ups. I agree with the reviewer's edits.

Two of the reviewers highlighted the issue of amino acid metabolism. I agree that your current wording is unclear and I do not understand what exactly you mean in the third paragraph of your discussion (the paragraph begins on line 364 with the words "Ants are a high-lipid, high-protein food..."). As written, it sounds like you did find *some* transcripts (but not many) encoding genes involved in biosynthesis of "...lysine, valine, leucine, isoleucine, phenylalanine, and tryptophan". I suspect that is wrong -- those amino acids are essential in humans, mice, dogs, cats, and cattle are therefore likely to be essential in mammals (I have not checked other vertebrates). I assume the authors intended to state that they *did not find any* transcripts encoding proteins necessary for the biosynthesis of essential amino acids.

This brings up the issue of arginine biosynthesis. I think transcriptomes are excellent for showing you which genes are present but no good for showing you which genes are absent. Indeed, I approach claims of gene absence with substantial skepticism, even if they arise from whole genome sequencing. Unless the genome assembly is excellent and the annotation is complete it is very easy to miss genes. It is trivially easy to miss genes in trascriptome data. It is possible that the arginine biosynthesis genes were present in the pangolin genome but repressed due to environmental conditions. I would strongly urge the authors to state that the absence of genes cannot be deduced from transcriptomes with certainty.

I hope these comments, along with those from the reviewers, are helpful as you move forward revising the manuscript. I did find the manuscript interesting and I am looking forward to the revision.

Reviewer 1 ·

Basic reporting

This manuscript provides a very thorough accounting of the data, methods, and results. The narrative could be improved by more clearly linking the transcriptomics approach with both evolutionary (e.g. comparisons with carnivores) and species management/conservation questions and implications.

The manuscript would benefit greatly from the assistance of a native english speaker to ensure correct grammatical usage and to make several parts clearer. There are a few spots where as written, it would appear that the correct meaning is not conveyed.

Experimental design

No comment.

Validity of the findings

By the nature of the study design, the comparisons of what is observed in ONE pangolin and only a certain number of tissues is compared against databases from multiple species and individuals and settings. Correlations that are highlighted need to done emphasizing that these point to areas where further research might be very useful for theoretical and practical purposes.

·

Basic reporting

This is a very interesting and informative manuscript dealing with the transcriptomics of an unusual and little-studied mammal. In general, the manuscript is fairly well-written, although the English grammar does require some attention and I have tried to point these instances out in the accompanying annotated document.

In general, the manuscript is well-composed, although there were numerous instances where statements were made but not supported by a suitable reference. Again, I have tried to highlight these instances in the accompanying document.

Experimental design

With specific reference to the study animal, I would like to see more information included on the specimen used. Ideally, the authors should include the mass and/or length of the study animal to give some indication of its age. Also, I would like to see more information on where this individual was sourced, specifically whether it was a captive-bred individual, a wild individual that was maintained in captivity (and if so, how long was it in captivity for?), or a completely wild individual. I believe that the origin of the individual could have a direct effect on the conclusions drawn from this study, as each scenario has different levels and origins of stress, and the diet of the individual (especially if it was sourced from captivity) could also affect the transcriptome classes recorded. These factors also need to be considered when comparing your results to those of Yusoff et al. (2016) – was their study animal also sourced from captivity?

Validity of the findings

I would like to see the results of this study compared and contrasted to the study by Yusoff et al. (2016), especially as this is the only other study (to the best of my knowledge) that has investigated transcriptomics in a pangolin species. Despite these studies looking at predominantly different organs (with the exception of the liver, which is common to both studies) I do feel that comparisons can be made, especially with regards to the proportion of transcripts that were assignable to each ontological class, as well as the proportion of transcripts assigned to each molecular function and cellular component class.

Additional comments

Please see the accompanying document for more specific comments.

This is a very interesting study that will definitely be a valuable addition to the growing body of literature on this threatened species, and I look forward to reviewing a revised version of this manuscript.

Reviewer 3 ·

Basic reporting

Minor grammatical errors occur in the manuscript. Some are listed in the 'General comments,' below. References to other studies that have successfully linked dietary requirements to transcript profiles would be useful, if they exist -- see 'General comments'.

Experimental design

The question was well-defined, and meaningful. The experimental design was constrained by the availability of samples -- biological replicates would presumably have been useful, but impossible to obtain.

Validity of the findings

In the General comments, I suggest some places where the links between the results and the conclusions ought to be bolstered. This might be achieved with the addition of references to studies in other species, or detailed explanations. Alternatively, greater qualifications might be added to the conclusions addressing uncertainties.

Additional comments

The manuscript by Ma et al. describes an analysis of transcriptomes from several tissues of the Malayan pangolin. Previously, no transcriptome has been completed for the salivary glands or intestine of this species. Here they were sequenced with the aim of providing information about the genes involved in myrmecophagy, and learning more about dietary requirements of this species, whose conservation might benefit from ex situ breeding. This motivation is outlined quite well in the introduction. The methods used to perform the transcriptome sequencing and assembly seem to me to be largely sound. I have a number of suggestions for the manuscript, but otherwise think it would make a worthwhile contribution to PeerJ.

Overall, my most serious comment is that I was sometimes uncertain that the results justified inferences that were drawn in the discussion. It is an important goal to look for hints about pangolin dietary requirements, but in some cases I think greater clarification, qualification, or references to relevant literature would be helpful. As an example, on Line 373, I found the section on the amino acids to be highly interesting. It suggests that the absence of transcripts for arginine synthesis might mean they are incapable of making this amino acid, so that arginine might be added to pangolin diets in captivity. (An aside: Line 376 seems to contradict this -- "These results suggested that M. javanica may synthesize arginine de novo." Wonder if this shoud say "not synthesise"?) However, I kept wondering, in the case of Arginine, what does the absence of transcripts mean, especially when the experiment was not replicated? How likely is it that transcripts involved in Arginine biosynthesis were present, but not observed due to low abundance, and random chance? Have studies in other species drawn correct (verified) inferences about dietary requirements based on transcriptomic profiling of different tissues? Could the available reference genome be interrogated more carefully to try to confirm the absence of genes, to corroborate this inference? Note, I do not suggest removing comments on this result from the discussion, which seems potentially informative, but it would be good to see it considered a more thoroughly, or qualified more carefully.


Moderately important points:

Line 145-146: Are there parameter values used for tophat that could be supplied to make the analysis more reproducible?

Line 154-157: Little information is provided about the databases, to explain why the transcripts are annotated using all of them. Do they offer complementary insights, or just a greater chance of locating a match for a given transcript that is observed?

Line 181-182: It is not clear what the "ratio of the transcripts" refers to...

Line 225-228: Terms such as 'cell' (Line 225) and 'cell part' are used without much explanation. For example, does 'cell' mean expressed in the cytosol?

Line 334: The analyses of differences in expression between tissues are interesting, but the results probably ought to be interpreted somewhat cautiously, since there is no (technical or biological) replication. Perhaps a qualification could be added to the methods or discussion.


Minor corrections / suggestions:

Throughout: Mohamad Yusoff et al. 2016 -> Yusoff et al. 2016?
From Yusoff et al. 2016: "How to cite this article: Yusoff, A. M. et al. De novo sequencing, assembly and analysis of eight different transcriptomes from the Malayan pangolin. Sci. Rep. 6, 28199; doi: 10.1038/srep28199 (2016)."

Line 43 myrmecophage -> myrmecophagy

Line 44 protection seems awkward here.

Line 48 eutherians and placentals seems redundant.. perhaps "are unique among plancental mammals"

Line 92 suggestion: "the last two years"

Line 97 "popping up like mushrooms" perhaps too informal?

---

## Round 0.2 · Minor Revisions

I am happy to accept this excellent manuscript pending one minor revision. My concern relates to a paragraph that was edited:

(this material begins on line 394 in the revised pdf)
"Few transcripts were involved in encoding proteins that are necessary for the biosynthesis of essential amino acids in humans, such as lysine, valine, leucine, isoleucine, phenylalanine, and tryptophan (Galili et al. 2016; Zhenyukh et al. 2017). These results suggest that M. javanica might have a limited ability to synthesize these amino acids, and arginine might be the first limiting amino acid. Therefore, it might be helpful to add these amino acids to manufactured feed. However, these points need further research, which might be very useful for theoretical and practical purposes."

I think I understand what you are saying, but I am still a bit confused. If I understand correctly, you are stating that you have genetic evidence that pangolins (or, at the very least, the Malayan pangolin) might be able to synthesize Lys, Val, Leu, Ile, Phe, and Trp. You have no evidence for Arg biosynthesis, although it is important to recognize (as you acknowledge by your use of "might") that the absence of Arg biosynthesis gene transcripts in transcriptome data does not provide evidence for their absence in the genome. Is that correct?

There is a part that appears garbled. By definition, we do not expect any human transcripts involved in biosynthesis of essential amino acids. So you should refer to the transcripts as encoding proteins involved in the biosynthesis of amino acids that are essential in humans.

The part that confuses me is the "Therefore, it might be helpful to add these amino acids to manufactured feed." If it is correct that Malayan pangolins can synthesize many amino acids essential to humans (Lys, Val, Leu, Ile, Phe, and Trp) then it would NOT be necessary to add them to feed.

The issue I can see is that one might wish to be conservative regarding the diet of an endangered animal when maintaining that animal in captivity. In other words, it is probably unwise to start changing diets unless there as evidence they don't work. Does this seem like a reasonable summary of the issue?

If so, I think you could rewrite the material above as follows:

"We did find a few transcripts encoding proteins that are necessary for the biosynthesis of amino acids that are essential in humans, such as lysine, valine, leucine, isoleucine, phenylalanine, and tryptophan (Galili et al. 2016; Zhenyukh et al. 2017). These results suggest that M. javanica might have at least a limited ability to synthesize those amino acids and that arginine might be the first limiting amino acid. Therefore, it might not be necessary to add the amino acids that pangolins have the potential to synthesis to manufactured feed. However, the regulation of the relevant transcripts is not fully characterized and the ability of pangolins to flourish without those amino acids is unclear. This issue needs further research, which might be very useful for theoretical and practical purposes."

Hopefully, I have captured your meaning. Feel free to contact me if you have questions and/or if I have simply misunderstood.

I don't think this should be a difficult revision and I hope you can take care of it quickly. As I said, if I can be of any assistance just let me know!

In addition, Reviewer 2 has also identified some edits for you - see below.

·

Basic reporting

This manuscript has been greatly improved since its original submission, and the authors are to be commended for this. I found the manuscript to be well-written, unambiguous and for the most part easy reading.

Experimental design

The methods and experimental design appear to be appropriate.

Validity of the findings

No comment.

Additional comments

I have a few minor suggested edits to improve the manuscript, which are detailed below:

Abstract
Line 43: Suggest deleting “of”. Sentence would read “Our study is the first to investigate the molecular mechanisms underlying myrmecophagy in M. javanica.”


Introduction
Lines 50 & 51. Gaudin et al. (2009) revised the taxonomy of pangolins, restricted the genus Manis to the four Asian species while assigning the two African tree pangolin species to the genus Phataginus and the two terrestrial African pangolin species to the genus Smutsia. These results have also been subsequently confirmed with molecular techniques (see for example du Toit et al. 2014), and this nomenclature has been adopted by most researchers and taxonomic groups, including the IUCN Pangolin Specialist Group. I therefore suggest that this taxonomy be adopted here as well.
[Gaudin, T.J., Emry, R.J. and Wible, J.R., 2009. The phylogeny of living and extinct pangolins (Mammalia, Pholidota) and associated taxa: A morphology based analysis. Journal of Mammalian Evolution 16(4): 235–305.]
[Du Toit, Z., Grobler, J.P., Kotzé, A., Jansen, R., Brettschneider, H. and Dalton, D.L., 2014. The complete mitochondrial genome of Temminck's ground pangolin (Smutsia temminckii; Smuts, 1832) and phylogenetic position of the Pholidota (Weber, 1904). Gene 551(1): 49–54]

Lines 55–58. Pangolins have been reported feeding on far more than four ant and one termite species, as is reflected in the articles that are cited. The four ant and one termite species listed all originate from the Pietersen et al. 2015 reference. I suggest that this sentence either be revised to reflect the full list of ant and termite species that pangolins are reported to feed on, or more preferably just state that pangolins feed exclusively on ants and termites (and keep the current references).

Line 73. Replace “the’ with “a”. The sentence will read “…habitat as a result of deforestation”


Results
Line 203. Remove “compared with the exons”


Line 361. You need to state where this “referred liver” originates from. I suggest including the "Yusoff et al. 2016" reference before “Figure S8”.

Line 367. What previous study does this refer to? Include the Yusoff et al. 2016 reference here as well.


Discussion & Conclusion
Line 378. Insert “the” between “between” and “small”. The sentence will read “between the small intestine and liver.”

Line 401. “Our” should not be capitalized.

Line 440. Remove the space in “amino -acid”

Lines 441 & 442. I suggest combining these two sentences to improve the flow. This can be done by combining the sentences to read “Malayan pangolins are endangered mammals for which captive breeding provides an opportunity to study the molecular mechanisms of myrmecophagy.”


Figures
Figure 3. Please write the full name (not just the abbreviation) of each tissue type in each of the axis legends.


Tables
Table 1. Replace “ID” with “Organ”

Table 2. Replace “ID” with “Organ”

---

## Round 0.3 · accepted · Accept

Thank you for completing one additional round of revisions. I am happy to accept this manuscript about these fascinating animals. I look forward to seeing this work in press!